# CONTROLLED LLM DECODING VIA DISCRETE AUTO-REGRESSIVE BIASING

**Patrick Pynadath, Ruqi Zhang**
Department of Computer Science
Purdue University
West Lafayette, Indiana, 47906, USA
`{ppynadat, ruqiz}@purdue.edu`

## ABSTRACT

Controlled text generation allows for enforcing user-defined constraints on large language model outputs, an increasingly important field as LLMs become more prevalent in everyday life. One common approach uses energy-based decoding, which defines a target distribution through an energy function that combines multiple constraints into a weighted average. However, these methods often struggle to balance fluency with constraint satisfaction, even with extensive tuning of the energy function's coefficients. In this paper, we identify that this suboptimal balance arises from sampling in continuous space rather than the natural discrete space of text tokens. To address this, we propose *Discrete Auto-regressive Biasing*, a controlled decoding algorithm that leverages gradients while operating entirely in the discrete text domain. Specifically, we introduce a new formulation for controlled text generation by defining a joint distribution over the generated sequence and an auxiliary bias sequence. To efficiently sample from this joint distribution, we propose a Langevin-within-Gibbs sampling algorithm using gradient-based discrete MCMC. Our method significantly improves constraint satisfaction while maintaining comparable or better fluency, all with even lower computational costs. We demonstrate the advantages of our controlled decoding method on sentiment control, language detoxification, and keyword-guided generation. We make our code available at the following url: `https://github.com/patrickpynadath1/dab`.

## 1 INTRODUCTION

Large language models (LLMs) are widely used in real-world applications through chatbots such as ChatGPT, Alpaca, and Llama, making them an important part of everyday life (Bender et al., 2021; Bommasani et al., 2021; Weidinger et al., 2021). As a result, there has been growing attention on developing methods to reliably and effectively control LLM-generated outputs to meet user-defined constraints (Gehman et al., 2020; Dathathri et al., 2020; Goshvadi et al., 2023; Han et al., 2024).

Previous work has tackled controlled language generation using decoding-time algorithms, which bypass the need for fine-tuning the base language model (Liu et al., 2023a; Kumar et al., 2022; Mireshghallah et al., 2022; Dathathri et al., 2020; Qin et al., 2022). Among these, energy-based decoding methods define a target distribution through an energy function, combining multiple constraints into a weighted average. This formulation offers significant flexibility, as the energy function can be any arbitrary function. Sampling from this distribution relies on gradient-based MCMC in continuous spaces, followed by conversion back to discrete text tokens.

However, these methods often struggle to achieve a good balance between fluency and constraint satisfaction. Tuning the coefficient for each constraint in the energy function requires exhaustive efforts, and even with careful tuning, the generated outputs may still fail to meet the desired standards.

In this paper, we analyze this issue and show that the suboptimal balance arises from sampling in continuous space rather than the natural discrete space of text tokens. Continuous-space sampling leads to incremental sequence updates, which hinders the exploration of fluent and constrained text.

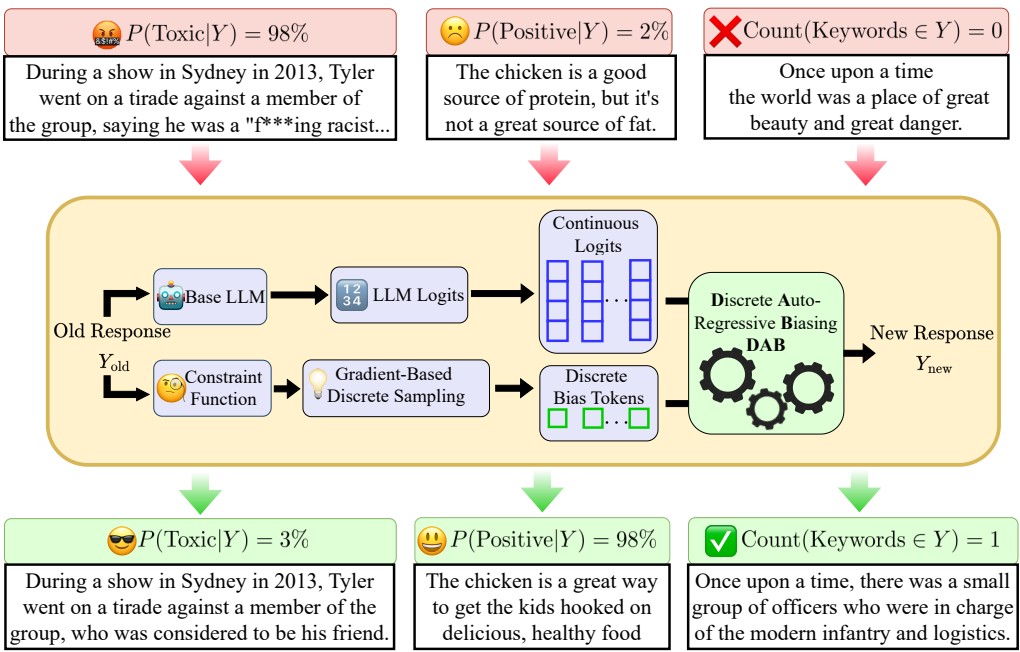

Figure 1: Visualization of our proposed controlled decoding algorithm, Discrete Auto-Regressive Biasing (DAB). Given an initial response that fails to satisfy some external constraint, DAB steers auto-regressive generation towards satisfactory generations using discrete bias tokens obtained via gradient-based discrete sampling from the constraint function.

To address this, we propose **DAB**: **D**iscrete **A**uto-regressive **B**iasing, a decoding algorithm that efficiently explores the discrete text space, achieving a better balance between fluency and constraint satisfaction with even lower decoding costs than existing decoding-time algorithms. DAB operates entirely in the discrete domain, which allows for more directional and substantial updates to the response sequence at each step. Specifically, our method defines a target distribution as a joint distribution over response sequences $Y$ and an auxiliary bias term $B$. It alternates between sampling from $P(B|Y)$ using gradient-based discrete sampling, ensuring constraint satisfaction, and from $P(Y|B)$ using biased auto-regressive generation, ensuring fluency. Notably, DAB significantly reduces computational overhead due to a simpler gradient computation which is enabled by our discrete sampling framework. We provide a visual summary of our algorithm in Figure 1. We summarize our main contributions as follows:

- We propose a controlled auto-regressive decoding algorithm that leverages gradients while operating entirely in the discrete text domain without continuous relaxation or post-hoc discretization. By remaining in the discrete domain, our method achieves a better balance between fluency and constraint satisfaction, with significantly lower computational costs than existing energy-based decoding algorithms.

- We introduce a new formulation for controlled text generation by defining a joint distribution over the generated sequence and an auxiliary bias sequence. To sample from this joint distribution, we propose a discrete Langevin-within-Gibbs sampling algorithm. Our algorithm leverages discrete gradient-based MCMC to sample the auxiliary bias sequence, which is then incorporated into the final output via biased auto-regressive generation.

- We demonstrate that our method consistently produces more satisfactory generations than baseline methods, offering comparable or better fluency with **2x** faster decoding time. This performance holds across a range of constrained generation tasks, including sentiment-controlled generation, toxicity avoidance, and keyword-guided generation.

## 2 RELATED WORK

### 2.1 LANGUAGE MODELS AS EBMS

Energy-based models (EBMs) have been a popular framework used to study the problem of inference-time controlled text generation. Initial works focused on the application of Gibbs sampling to encoder-based architectures such as BERT (Wang & Cho, 2019; Goyal et al., 2022; Mireshghallah et al., 2022).

While these methods are similar to ours in that they perform discrete sampling, our work differs in several key ways: (1) Our work leverages gradient information for a more informed exploration of the energy landscape. (2) Our Gibbs sampling alternates between the response and bias, updating the entire response at once. whereas previous Gibbs sampling approaches only update a single token at a time. (3) Our method leverages auto-regressive generation within decoder architectures, whereas these works can only be used with encoder architectures.

More recent works have applied gradient-based sampling methods to more efficiently navigate the energy landscape. Qin et al. (2022) that uses Langevin dynamics in the logit space followed by top-k masking; Kumar et al. (2022) uses Langevin dynamics in the embedding space followed by a projection to the embedding space; and Liu et al. (2023a) uses the ADAM optimizer with Gaussian noise to obtain bias terms for biased auto-regressive generation.

Our approach differs from these works in that it relies on discrete sampling rather than continuous sampling or optimization. By operating directly in the natural discrete space of text tokens, our method not only improves the trade-off between fluency and constraint satisfaction but also reduces decoding costs.

Beyond MCMC methods, it is also possible to apply some modified rejection sampling to proposal algorithm to improve approximate sampling of the target EBM (Eikema et al., 2022). Additionally, there is also a line of research devoted to fine-tuning LLMs to align with an EBM defined in terms of the base LM and external constraints (Khalifa et al., 2020; Korbak et al., 2022; Kruszewski et al., 2023). We choose to focus on inference-time algorithms as they provide more flexibility and avoid long training runs.

### 2.2 ALTERNATIVE CONTROLLED GENERATION APPROACHES

While the EBM framework is popular within the field of controlled text generation, there are several alternative approaches. In terms of previous inference time algorithms, many works rely on specially trained auxiliary models to provide token-level guidance (Krause et al., 2020; Yang & Klein, 2021; Liu et al., 2021; Meng et al., 2022; Kim et al., 2022) or query a standard text classifier multiple times per generated token (Dekoninck et al., 2024; Sitdikov et al., 2022). Other works apply gradient-based optimization methods to improve constraint satisfaction (Qin et al., 2020; Dathathri et al., 2020).

Recently, Han et al. (2024) introduces LM-Steer, a method for learning linear transformations to influence language model generation. This is similar to the BOLT algorithm Liu et al. (2023a) as both alter the embedding representations from auto-regressive generation by some learned operation. Whereas Liu et al. (2023a) is framed as a decoding-time algorithm, LM-Steer requires training data in order to learn the linear transformation. In general, these methods either suffer from the steep trade-off between fluency and control previously mentioned, require separate training or fine-tuning for the constraint LM, or are computationally expensive.

### 2.3 GRADIENT-BASED DISCRETE SAMPLING

Recent works have demonstrated the benefits of leveraging gradient information for sampling over discrete spaces (Grathwohl et al., 2021; Sun et al., 2023a; Goshvadi et al., 2023; Sun et al., 2023b; Pynadath et al., 2024). Our method uses the sampling algorithm introduced in Zhang et al. (2022), which can be viewed as the analog of Langevin dynamics (Roberts & Stramer, 2002) in discrete spaces. Our paper may be of interest to this field as it is the first to explicitly link discrete gradient-based sampling with controlled language model generation.

## 3 PRELIMINARIES

**Task Definition**   Let $P^{LM}$ denote some pre-trained language model, $V$ denote the set of tokens in the model vocabulary, and $f : |V|^n \to \mathbb{R}$ represent an external constraint where higher values correspond to better constraint satisfaction. We will use $Y = \{y_1, y_2 \ldots y_n\}$ to refer to the sequence of tokens in the response.

We assume $P^{LM}$ is an auto-regressive transformer as used in previous work (Qin et al., 2022; Kumar et al., 2022; Liu et al., 2023a). Given some initial prompt $X$, we define the auto-regressive distribution for the $i$th position as $\tilde{y}_i = P^{LM}(\cdot|y_{<i}, X)$. This forms a distribution over the vocabulary $V$ and can be represented as a $|V|$ dimensional logit vector.

We define the task of controlled language generation as generating a sequence of $n$ tokens $Y = \{y_1, y_2, \ldots y_n\}$ from some initial prompt $X = \{x_1, x_2 \ldots x_m\}$ that is both high in likelihood under the language model and high in terms of constraint satisfaction. We can compute the likelihood of the generation under the pre-trained language model for a sequence of length $n$ as follows:

$$P^{LM}(Y|X) = \Pi_{i<n} \, P^{LM}(y_i|y_{<i}, X). \tag{1}$$

Previous works have framed this problem as sampling from an unnormalized distribution, commonly referred to as an energy based model (EBM) (Mireshghallah et al., 2022; Qin et al., 2022; Kumar et al., 2022; Liu et al., 2023a). The *energy function* defines an unnormalized distribution and is typically defined as follows:

$$E(Y) = \lambda_1 \log P^{LM}(Y|X) + \lambda_2 f(Y|X). \tag{2}$$

**Non Auto-regressive Generation**   As the denominator, or partition function, requires computing the energy of all possible sequences, it is intractable to directly sample from $\pi$. Previous works address this by applying Langevin dynamics as it only requires gradients of the energy function $E$ (Qin et al., 2022; Kumar et al., 2022). Specifically, they use some continuous representation of the current sample $\tilde{Y}_t$ and a learning rate $\gamma$ to define the following update step:

$$\tilde{Y}_{t+1} = \tilde{Y}_t + \gamma \nabla_{\tilde{Y}} E(\tilde{Y}_t) + \epsilon, \epsilon \sim \mathcal{N}(0, \sigma^2 I) \tag{3}$$

Non-autoregressive generation methods typically rely on some form of filtering or projection to ensure that the continuous generation can be mapped back into the discrete token space $V$. In Qin et al. (2022), the final generation is filtered using a top-k mask, where the top-k indices are obtained from the base language model $P^{LM}$. In Kumar et al. (2022), their proposed algorithm performs the update in the embedding space and projects the resulting vector onto the set of token embeddings for the base language model.

**Biased Auto-regressive Generation**   Liu et al. (2023a) introduced a method that samples a bias term from the target distribution and incorporates it into auto-regressive generation. While the sampling step of the bias-term is similar to equation 3 in the embedding space, they skip the projection step of Kumar et al. (2022) and modify the auto-regressive step as follows:

$$y_i = \underset{j \in |V|}{\arg\max} \left( \tilde{y}_{i,j} + w_i \cdot (b_i M^T)_j \right) \tag{4}$$

Here, $i$ is the sequence position, $\tilde{y}_i$ is the initial logit vector, $M$ is the embedding table for the language model $P^{LM}$, $b_i$ is the bias vector in the embedding space, $w_i$ is the weight value, and $j$ refers to the logit coordinate. $b_i M^T$ refers to the transformation of the bias vector $b_i$ to a logit vector. Liu et al. (2023a) demonstrates that sampling a bias term to direct auto-regressive generation enables quicker convergence to satisfactory generation, as opposed to non auto-regressive generation. However, they also mention the undesirable trade-off between control towards constraint satisfaction and fluency, which our work addresses.

## 4 DISCRETE AUTOREGRESSIVE BIASING

In this section, we introduce DAB: **D**iscrete **A**uto-regressive **B**iasing. First, we present the formulation of the target distribution as a joint distribution and explain the motivation behind this approach.

Next, we describe how our algorithm samples from the joint distribution by alternating between biased auto-regressive generation and discrete gradient-based sampling. We finally demonstrate that gradient-based discrete sampling enables our algorithm to have more thorough, stable, and efficient sampling when compared to continuous methods.

## 4.1 FORMULATION

Our goal is to formulate the target distribution in such a way that enables both auto-regressive biasing and discrete sampling, allowing for superior exploration of the discrete space of fluent and controlled responses. However, existing formulations will not allow for both: while Kumar et al. (2022) introduces a framework that allows for direct sampling of word embedding, it is non auto-regressive; and while Liu et al. (2023a) introduces a method that allows for auto-regressive generation, the bias vectors used are continuous. Thus here we introduce a new formulation.

**Target Discrete Distribution**  In order to ensure the constraint satisfaction of the output sequence $Y$, we introduce an auxiliary variable $B = \{b_1, b_2 \ldots b_n\}$, where each $b_i \in V$. We refer to this sequence as the bias sequence or bias tokens. First, we define the joint distribution over $Y, B$ conditioned on the prompt $X$:

$$P(Y, B|X) \propto P^{LM}(Y|X, B) \exp(f(B|X)). \tag{5}$$

Here, $f(B|X)$ represents the constraint, with larger values of $f(B|X)$ indicating better satisfaction of the constraint. $P^{LM}$ refers to the language model distribution conditioned on $X, B$.

**Marginal Distribution** $P(Y|X)$  The marginal distribution of $Y$ is:

$$P(Y|X) = \sum_{B \in |V|^d} P(Y|X, B) P(B|X) = \sum_{B \in |V|^d} P(Y|X, B) \frac{\exp(f(B|X))}{Z_B}.$$

This formulation expresses the probability of obtaining the response $Y$, taking into account all possible biases $B$. The response $Y$ drawn from this marginal distribution will be both fluent (due to the term $P^{LM}(Y|X, B)$) and highly satisfying of the constraints (due to $\exp(f(B|X))$). The bias variable $B$ helps balance these two aspects. As the distribution over $Y$ incorporates both $P^{LM}$ and the external constraint, probable sequences under this distribution will be both fluent and satisfactory. This ensures that the generated response $Y$ from our algorithm has the desired properties.

**Marginal Distribution** $P(B|X)$  The marginal distribution of $B$ is given by:

$$p(B|X) = \frac{\exp(f(B|X))}{Z_B} \tag{6}$$

This indicates that highly probable values of $B$ will satisfy the external constraint. However, since the distribution does not incorporate the language model $P^{LM}$, the sequence of $B$ may not be fluent. For this reason, we use $B$ as a "guide" sequence as opposed to using it as the response.

**Conditional Distributions**  Given the joint distribution defined in equation 5, the conditional distribution of $Y$ is $P(Y|B, X) = P^{LM}(Y|X, B)$. Furthermore, the conditional distribution of $B$ is:

$$P(B|X, Y) = \frac{P^{LM}(Y|X, B) \exp(f(B|X))}{P(Y|X)} \propto P^{LM}(Y|X, B) \exp(f(B|X))$$

**Advantages of Joint Distribution**  The primary motivation behind our framework is the observation that fluency is best satisfied through auto-regressive generation, and gradient-based sampling efficiently finds responses that satisfy constraints. By framing the problem as a joint distribution of $Y, B$, we enable the use of both methods. The typical target shown in equation equation 2 does not support auto-regressive generation and significantly compromises fluency.

## 4.2 Sampling Algorithm

Sampling directly from the joint distributions of $Y, B$ is challenging. We propose to use Gibbs sampling to sample from the target distribution by alternatively sampling from $P(Y|X,B)$ and $P(B|X,Y)$. First, we describe how our proposed algorithm samples from each conditional distribution. We then introduce the complete sampling algorithm along with some intuition as to how it ensures satisfactory and fluent generations.

**Sampling from $P(B|X,Y)$** The conditional distribution for $P(B|X,Y)$ includes the term $P^{LM}(Y|X,B)$. This is to ensure that the sampled $B$ results in output sequences that are high in likelihood under the base model distribution. However, directly computing $P^{LM}(Y|X,B)$ for all possible values of $B$ is intractable. By noting that this term encourages the selection of $B$ that is consistent with the observed $Y$, we can infer this property is naturally satisfied when $B$ is close to $Y$. Thus, we approximate $P^{LM}(Y|X,B)$ by performing a single MCMC step with the initial state set to $Y$.

In order to sample this discrete sequence of tokens, we apply the Discrete Langevin Proposal (DLP) introduced in (Zhang et al., 2022). After initializing $B$ as $B = Y$, and representing the sequence as a sequence of one-hot vectors $\hat{B} = \{\hat{b}_1, \hat{b}_2 \ldots \hat{b}_n\}$, we execute a single step of DLP with the target distribution being $\exp(f(B|X))$. Below we include the proposal distribution for position $i$:

$$b'_i \sim \text{Categorical}\left(\underset{j \in |V|}{\text{softmax}}\left(\frac{1}{\tau}(\nabla f(\hat{B}|X))_{i,j}(1 - \hat{b}_{i,j})\right)\right) \tag{7}$$

Here, $\tau$ is a temperature hyper-parameter that controls the sharpness of the proposal distribution, $(\nabla f(\hat{B}|X))_{i,j}$ is the $j$th component of the $i$th gradient vector, $\hat{b}_{i,j}$ represents the $j$th component of the one-hot vector $\hat{b}_i$, and $b'_i$ is the token we sample from the distribution over $V$. For more details on the application of DLP and the gradient compution, see Appendix A. Unlike the algorithm presented in Liu et al. (2023a), we do not require the use of straight through estimation (STE) (Bengio et al., 2013) as we differentiate directly with respect to $Y$. Note that this proposal function can be computed for all sequence positions in parallel. We refer to this proposal function as $q_\tau(\cdot|B)$.

**Sampling from $P(Y|X,B)$** Our goal is to sample from $P(Y|X,B)$ using biased auto-regressive generation, similar to equation 4. In order to do so, we must map the sequence of bias *tokens* $B$ to a sequence of bias *vectors* $\tilde{B}$. The ideal bias vector should reflect the difference in meaning between each token in the vocabulary space $V$ and the sampled token. To accomplish this, we penalize each token based on the distance to the sampled bias token within the embedding space, as static embeddings reflect semantic meaning (Mikolov, 2013; Pennington et al., 2014; Mikolov et al., 2013). Given a bias token $b_i$, embedding table $M$, we define the $j$th coordinate value corresponding to token $v_j$ as follows:

$$\tilde{b}_{i,j} = ||Mb_i - Mv_j||_2^2 \tag{8}$$

This yields a $|V|$ dimensional vector that can be added to the auto-regressive logits $\tilde{y}_i$. When adding the bias term to $\tilde{y}_i$, we also incorporate both a weight term $w_i$ and a normalizing factor $r_i$. While $w_i$ is a hyper-parameter, $r_i$ normalizes the bias vector at the $i$ position to have the same norm as $\tilde{y}_i$. We define the normalizing factor as follows:

$$r_i = \frac{||\tilde{y}_i||_2}{||\tilde{b}_i||_2} \tag{9}$$

We note that while this normalizing factor can also be applied to BOLT and may improve its results, the modified BOLT still underperforms compared to our method.

We formally define our biased auto-regressive generation as follows:

$$y_i = \underset{j \in |V|}{\arg\max}\left(\tilde{y}_{i,j} - w_i \cdot r_i \cdot \tilde{b}_{i,j}\right). \tag{10}$$

Intuitively, this returns the token corresponding to the maximal coordinate of the biased distribution. Repeating this $n$ times results in the updated response sequence $Y$.

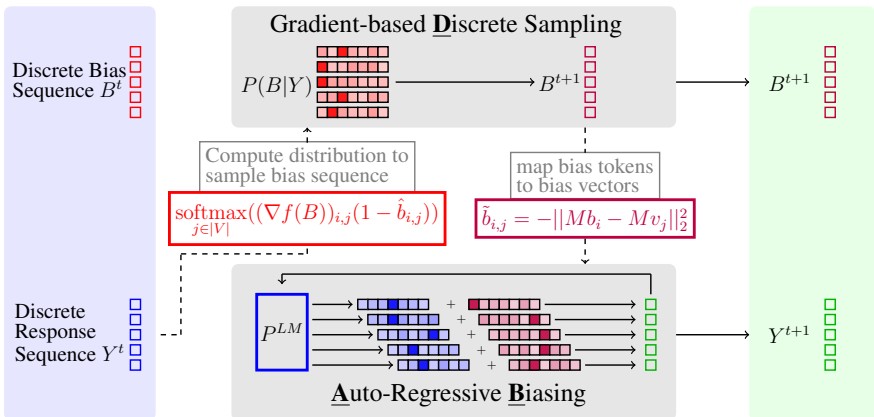

Figure 2: Visualization of the proposed decoding algorithm, DAB. DAB alternates between sampling the response $Y$ and the bias $B$. To sample $B$ given $Y$, we use gradient-based discrete sampling on the constraint function $f$. To sample $Y$ given $B$, we compute a bias vector that penalizes words based on their distance to $B$ and then use this bias to guide the auto-regressive generation.

**Text Generation Algorithm** We provide a visualization of our algorithm in Figure 2, and include the full algorithm in Appendix B. Given some prompt, we first generate some initial auto-regressive generation $Y_1$, with the initial bias vector set to $\vec{0}$. After obtaining $Y$, we sample from the conditional distribution over $B$ to obtain a sequence of bias tokens. We then use equation 8 to compute the new bias vector to use for biased auto-regressive generation. We repeat this alternative sampling process for several iterations, returning the sample that best satisfies the constraint at the end as commonly done in the literature (Kumar et al., 2022; Liu et al., 2023a). For a discussion on the hyper-parameters of our algorithm, see Appendix C.

## 4.3 Advantages of Biasing in Discrete Spaces

Here we discuss various advantages of discrete sampling in the context of auto-regressive biasing. First, we demonstrate that discrete sampling enables a quicker and more thorough exploration of potential output sequences $Y$. We then describe how discrete sampling solves the stability issue discussed in Liu et al. (2023a). Finally, we show that discrete sampling makes use of simpler gradient computations, resulting in a more efficient decoding algorithm.

**Exploration of State Space** Discrete sampling enables DAB to explore the output space more effectively than continuous methods. We hypothesize that discrete sampling enables more directional and substantial changes to the bias vector, resulting in more token changes in the output sequence across sampling steps. We compare with BOLT, a continuous auto-regressive biasing algorithm (Liu et al., 2023a). We examine the performance of BOLT both with and without the normalizing factor defined in equation 9. We include the comparison of hops across 50 steps in Figure 3a. These results show that our method updates substantially more sequence positions across all sampling steps than either variant of BOLT.

Next, we measure how comprehensively each method explores the sample space of potential sequences. For each sequence position, we maintain a record of tokens encountered throughout the sampling process and compute the number of unique tokens within this set. Figure 3b shows the average unique tokens per sequence position for all three algorithms. These results indicate that our method samples more unique tokens for each sequence position than either variant of BOLT, demonstrating more comprehensive exploration. Collectively, these findings confirm that discrete sampling enables faster, more thorough, and thus more effective exploration of the sample space of potential sequences.

**Sampling Stability** Discrete sampling allows DAB to have superior stability across sampling steps when compared to continuous methods. We show this in Figure 3c, where we track the average

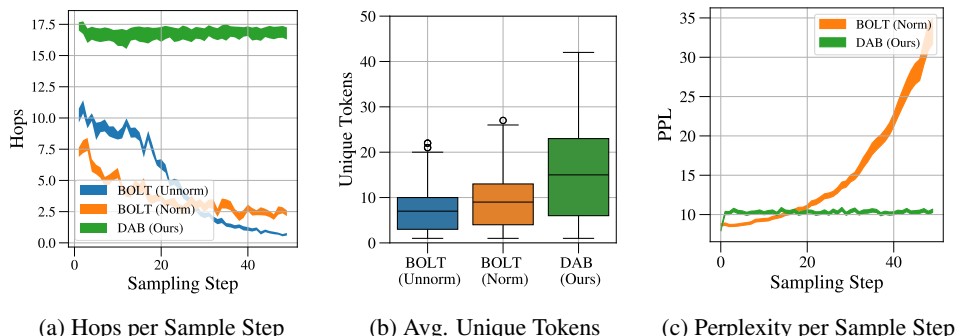

(a) Hops per Sample Step    (b) Avg. Unique Tokens    (c) Perplexity per Sample Step

Figure 3: (a) Average hops, or token updates per sequence, against sampling steps. Both versions of BOLT suffer from decreasing hops while DAB remains stable. (b) Average number of the unique tokens sampled for each sequence position throughout the entire sampling process. DAB discovers many more unique tokens for each position than either variant of BOLT. (c) Comparison of fluency with respect to sampling steps. Dab exhibits stable fluency over sampling steps in comparison to BOLT.

perplexity of the batch at each time step. While BOLT faces deteriorating perplexity, DAB remains stable throughout the sampling process.

We attribute this instability to the difficulty of applying continuous sampling techniques to a discrete domain as discussed in Grathwohl et al. (2021). As a result of BOLT's misalignment between the sampling domain and target domain, the energy landscape is too complex to navigate with gradient information. This results in the divergence seen in Figure 3c.

Our algorithm avoids this entirely as we perform direct sampling on the discrete token space. Since we define the sampling domain and target domain to be the same, our algorithm enjoys superior stability throughout all sampling steps. This improvement in algorithmic stability removes the need for implementing early-stopping and to carefully tune the number of sampling steps.

Table 1: Efficiency comparison between BOLT and DAB in terms of tokens per second.

|  | BOLT | DAB (Ours) |
|---|---|---|
| Tokens per Second | $9.495 \pm .095$ | $\mathbf{23.213 \pm 0.304}$ |

**Improved Efficiency**  Discrete sampling enables our algorithm to use simpler gradient computations that provide a computational advantage over continuous sampling methods. To evaluate our algorithm's efficiency, we compare the tokens per second of our method to BOLT.

We also measure the time-cost for computing the bias term for each method. We put the results in Table 1, where we observe that our method has over **2x** the tokens per second output when compared against BOLT. Our algorithm achieves this computational advantage as a result of computing the gradient with respect to $\hat{B} = \hat{Y}$, which removes the need to backpropagate through auto-regressive generation. Computing the gradient of $f$ with respect to a continuous bias term $\tilde{B}$ requires first computing $\partial f / \partial \hat{Y}$ and then $\partial \hat{Y} / \partial \tilde{B}$. Since each one-hot vector in $\hat{Y}$ is influenced by previous bias terms, the latter term requires backpropagation through auto-regressive generation. Simply initializing $\tilde{B} = \hat{Y}$ will not work in continuous sampling because the incremental updates will keep $\tilde{B}$ close to the original $\hat{Y}$. In contrast, our method uses gradients to identify which tokens will increase constraint satisfaction and directly samples them, enabling substantial change from the original sequence while incorporating information from the external constraint. While continuous sampling cannot exploit this computational shortcut and maintain constraint satisfaction, gradient-based discrete sampling achieves both simultaneously.

## 5 EXPERIMENTS

**Tasks**  We evaluate DAB on three distinct controlled-generation tasks: sentiment-guided generation, language detoxification, and keyword-guided generation. These are popular tasks within the field of controlled generation (Kumar et al., 2022; Liu et al., 2023a; Han et al., 2024). For all tasks,

Table 2: Sampling algorithm performance on sentiment-directed generation, language detoxification, and keyword-guided generation. DAB acheives superior control metrics than baselines across all task. while also demonstrating comparable or better fluency metrics competitive with the best baseline.

| Sentiment | Control | | | Fluency | | |
|---|---|---|---|---|---|---|
| | *Int. Clsf* ↑ | *Ext. Clsf (Yelp)* ↑ | *Ext. Clsf (SST-2)* ↑ | *CoLA* ↑ | *REP-3gram* ↓ | *PPL* ↓ |
| MuCOLA | .841 ± .009 | .843 ± .011 | .899 ± .008 | .681 ± .008 | .091 ± .006 | 34.786 ± 2.205 |
| COLD | .697 ± .011 | .515 ± .015 | .670 ± .013 | .731 ± .008 | .061 ± .003 | 15.908 ± .394 |
| BOLT | .903 ± .006 | .747 ± .013 | .878 ± .001 | **.874 ± .005** | **.0008 ± .0002** | **9.919 ± .142** |
| LM-Steer | - | **.900 ± .008** | .948 ± .006 | .564 ± .008 | .117 ± .007 | 72.153 ± 3.195 |
| DAB *(Ours)* | **.992 ± .001** | .894 ± .009 | **.975 ± .003** | .860 ± .005 | .004 ± .001 | 11.773 ± .203 |

| Toxicity | *Int. Clsf* ↓ | *Avg. Max Toxicity* ↓ | *Toxicity Pred. Prob.* ↓ | *CoLA* ↑ | *REP-3gram* ↓ | *PPL* ↓ |
|---|---|---|---|---|---|---|
| MuCOLA | .098 ± .002 | .269 ± .006 | 7.6% | .691 ± .002 | .006 ± .001 | 58.015 ± .435 |
| COLD | .136 ± .002 | .266 ± .007 | 10.2% | .667 ± .001 | .024 ± .001 | 38.891 ± .177 |
| BOLT | .065 ± .001 | .264 ± .006 | **6.8%** | **.830 ± .001** | .001 ± .0001 | 27.283 ± 2.233 |
| LM-Steer | - | .265 ± .006 | 7.9% | .722 ± .002 | .006 ± .002 | 52.697 ± .356 |
| DAB *(Ours)* | **.057 ± .001** | **.211 ± .006** | **6.8%** | .806 ± .001 | .001 ± .0001 | **25.609 ± .126** |

| Keyword | *BertScore* ↑ | *Success Rate* ↑ | - | *CoLA* ↑ | *REP-3gram* ↓ | *PPL* ↓ |
|---|---|---|---|---|---|---|
| MuCOLA | .8083 ± .0004 | **100%** | - | .248 ± .004 | .007 ± .001 | 475.301 ± 30.445 |
| COLD | .8123 ± .0005 | **100%** | - | .205 ± .003 | .020 ± .001 | 241.980 ± 4.943 |
| BOLT | .8291 ± .0003 | 99.1% | - | .705 ± .006 | .005 ± .005 | 32.019 ± 1.593 |
| DAB *(Ours)* | **.8303 ± .0003** | 99.0% | - | **.726 ± .005** | **.004 ± .001** | **23.424 ± .317** |

we produce generations in batches of 20 as done in Liu et al. (2023a). For all tasks, we include example generations in Appendix D.2, D.3, D.4 for sentiment directed generation, language detoxification, and keyword-guided generation respectively.

**Baselines** We compare to previous generation algorithms that use the EBM framework to perform gradient-based text sampling. Specifically, we compare to MuCOLA (Kumar et al., 2022), COLD (Qin et al., 2022), and BOLT (Liu et al., 2023a). We also compare against LM-Steer introduced in (Han et al., 2024) to see how our method compares to alternative controlled generation methods.

**Metrics** While the metrics assessing control towards external constraint vary across experiments, we use the same evaluation metrics to measure fluency across experiments. We measure fluency by looking at CoLA score, the number of repeated tri-grams per generation, and perplexity (Kumar et al., 2022; Liu et al., 2023a). For CoLA score, we use a fine-tuned RoBERTa model to provide a probability as to whether a generation is grammatically correct. For perplexity, we use GPT-XL to evaluate each generation. We show the average of the results across all generations. For more details on these evaluation methods, refer to Appendix D.1.

## 5.1 SENTIMENT-CONTROLLED GENERATION

**Task** Here we measure the ability to direct generation towards positive sentiment from some initial prompt. Given an initial prompt, we generate sequences of length 12, 20, 50 as done in prior works (Kumar et al., 2022; Liu et al., 2023a). We use the same set of prompts as Dathathri et al. (2020) and include them in Appendix D.2.

**Control Metrics** We evaluate control by measuring the predicted sentiment of the generation using three distinct sentiment classifiers. For details on the training of the three classifiers, see Appendix D.2. We omit the internal classifier measure for LM-Steer as it does not rely on an internal classifier to guide generation.

**Results** Table 2 shows that our method achieves a better balance between control and fluency than baselines. DAB achieves the highest average probability of positive sentiment across all three classifiers, demonstrating its effectiveness at incorporating the external constraint. Furthermore, DAB achieves fluency scores close to BOLT's performance in regards to CoLA score, repeated trigrams, and perplexity. This shows that DAB produces generations that are both fluent and satisfactory under the constraint.

## 5.2 TOXICITY AVOIDANCE

**Task**  We compare our algorithm to various baselines for the task of language detoxification to demonstrate that our method can be used to mitigate potentially toxic LLM generations. Following prior work, we use 1,000 prompts sampled from the RealToxicityPrompts introduced and generate continuations of length 20 tokens (Gehman et al., 2020; Kumar et al., 2022; Liu et al., 2023a). As in the sentiment-directed generation task, we use a fine-tuned RoBERTa as the constraint function.

**Control Metrics**  We evaluate the generations using both the internal discriminator used to guide the various methods, and the score returned by the Perspective API (Lees et al., 2022). We use the scores returned from Perspective API to calculate the maximum toxicity per prompt and the overall percentage of text predicted to be toxic.

**Results**  As shown in Table 2, our method generates less toxic text than baselines without compromising fluency. DAB significantly decreases the average maximum toxicity per prompt, demonstrating that our algorithm is more consistent in terms of toxicity reduction. Furthermore, our method obtains fluency metrics that are on par with the best baseline.

## 5.3 KEYWORD-GUIDED GENERATION

**Task**  We measure the ability of our method to produce text that includes a given keyword relevant to a specific topic. As done in prior work, we use 7 topics with 4 keywords each and use the differentiable BLEU (Liu et al., 2022) as the constraint function (Liu et al., 2023a). For baselines, we compare to the same methods as before with the exclusion of LM-Steer as no similar task was discussed in the original work.

**Control Metrics**  The ideal metric goal for this task should only assign good scores to text where keywords are used in a meaningful way. While Liu et al. (2023a) uses the percentage of generations that include a keyword to measure constraint satisfaction, this metric also assigns good scores to text where the keyword does not add meaning to the sentence.

An alternative approach is to compute a similarity score between the generations and reference texts that use the keywords in a manner relevant to the given topic. To accomplish this, we use GPT-4o to produce sentences for each combination of prompt, topic, and keyword. We then use BertScore to compute the similarity score between the candidate generations and the reference text, where higher similarity implies increased relevance to the target topic (Zhang et al., 2020). For further details, refer to Appendix D.4.

**Results**  DAB is able to outperform baselines in terms of both control and fluency, reflecting an improved balance between these two attributes. As shown in Table 2, DAB outperforms baselines in terms of relevance towards the target topic as measured by BertScore. While the success rate of our method is not as high as some of the baselines, those methods do not incorporate the keywords in a semantically meaningful manner as indicated by the BertScore. Furthermore, our generations are more fluent than all baselines in terms of all three fluency metrics. These results indicate that our method strikes a superior balance between control and fluency in regards to topic-guided generation.

## 6 CONCLUSION

In this work, we introduce DAB, the first controlled decoding algorithm based on gradient discrete sampling. Our algorithm alternates between biased auto-regressive generation and gradient-based discrete sampling to produce text from pre-trained language models subject to an external constraint function. Through various controlled generation experiments, we demonstrate that our algorithm is both more efficient and effective than previous methods.

**Limitations**  While our method is more efficient than other gradient-based controlled generation methods, the number of gradient computations increases with queries, which may be undesirable. Furthermore, it has not yet been explored how our method performs when faced with multiple external constraints or compositional generation.

## Ethics

We adhere to the ICLR Code of Ethics. Additionally we confirm that our experiments use only public datasets. The algorithm introduced in this work is a general-purpose algorithm for directing LLMs to generate text satisfying arbitrary constraints. Thus it is possible to define malicious constraints that cause LLMs to produce harmful text. Similar to the work done in (Guo et al., 2024), it may be possible to apply our algorithm towards jail-breaking LLMs and causing them to produce harmful text. Previous works have demonstrated that it is possible to induce harmful behavior in LLMs via various attacks (He et al., 2024; Liu et al., 2023b; Schwinn et al., 2023).

## Reproducibility

In order to ensure the reproducibility of our work, we include the details necessary to replicate both the core algorithms and the experiments. We include the psuedo-code for our algorithm in 1; information on hyper-parameter settings in Appendix Cl and further details on each experiment in Appendix D.2, D.3, D.4. Additionally, we include the code-base used to produce our results at the following repository: https://github.com/patrickpynadath1/dab.

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

# A  DISCRETE LANGEVIN PROPOSAL

Our proposed controlled text generation leverages the gradient-based discrete sampling algorithm in Zhang et al. (2022), which is further investigated by Pynadath et al. (2024). Using the same notation as in the Main Body of the paper, we put the original proposal distribution from Zhang et al. (2022) below:

$$\text{Categorical}\left(\operatorname*{softmax}_{j\in|V|}\left(\frac{1}{2}\nabla f(\hat{B}|X)_i(\text{Onehot}_j - \hat{b}_i) - \frac{||\text{Onehot}_j - \hat{b}_i||_2^2}{2\alpha}\right)\right)$$

Here, $\hat{b}_i$ corresponds to the one-hot vector in sequence position $i$. Similarly, $\text{Onehot}_j$ corresponds to the one-hot vector for the $j$th token in $V$. This proposal function defines a distribution over the vocabulary for the $i$th sequence position in the sequence by taking the softmax over all possible tokens.

As discussed in Pynadath et al. (2024), this proposal is locally balanced, or optimal for very small step-sizes. For the task of controlled text generation, we would prefer a proposal function that is optimal for large step-sizes, which allow for superior exploration of the space of potential sequences. The globally balanced proposal can be written as follows:

$$\text{Categorical}\left(\operatorname*{softmax}_{j\in|V|}\left(\nabla f(\hat{B}|X)_i(\text{Onehot}_j - \hat{b}_i)\right)\right)$$

In terms of the gradient computation, the one-hot representation enables the use of automatic differentiation packages to compute $\nabla f(\hat{B}|X)$. We observe that the term $(\text{Onehot}_j - \hat{b}_i)$ corresponds to the distance between the proposed token $j$ and the original token $b_i$. We choose to represent this distance term as hamming distance, given the discrete nature of the space we wish to sample. For a token $j$, the hamming distance to the original token in position $i$ is 0 if the $j$th coordinate $\hat{b}_{ij} = 1$ as they are the same token; and 1 if the $j$th coordinate is 0. Thus we can represent the distances between the tokens as $1 - \hat{b}_{ij}$. This leads us to the proposal function in 7, which we place below for convenience:

$$b_i' \sim \text{Categorical}\left(\operatorname*{softmax}_{j\in V}\left(\frac{1}{\tau}(\nabla f(\hat{B}|X))_{ij}(1 - \hat{b}_{ij})\right)\right)$$

Here, $b_i'$ refers to the token we sample from the categorical distribution over $V$.

# B  ALGORITHMIC DETAILS

Here we provide the full pseudo-code for our algorithm.

---
**Algorithm 1** Discrete Autoregressive Biasing

---
**Require:** Constraint function $f$, $P^{LM}$, prompt $X$, number steps $s$, sequence length $n$, embedding table $M$
1: $\tilde{B} \leftarrow \vec{0}, f_{\min} \leftarrow -\infty, Y_{\text{best}} \leftarrow \{\}$ ▷ *Initialize constraint violation as being maximal and current best generation as empty*
2: **for** step $s$ **do**
3:    **for** position $i$ in range($n$) **do**
4:       $\tilde{y}_i \leftarrow \log P^{LM}(\cdot|y_{<i}, X)$ ▷ *Initial auto-regressive distribution over V*
5:       Calculate normalizing factor $r_i$ if $s > 1$, else $r_i \leftarrow 1$
6:       $y_i \leftarrow \operatorname{argmax}_{j\in|V|}\left(\tilde{y}_{i,j} - w_i \cdot r_i \cdot \tilde{b}_{i,j}\right)$ ▷ *Sample from $P(Y|X, B)$*
7:    **end for**
8:    $B \leftarrow Y$ ▷ *Initialize B as Y*
9:    Evaluate $f(B|X)$, update $f_{\min}$, $Y_{\text{best}}$
10:   $B' \sim q_\tau(\cdot|B)$ as in equation 7 ▷ *Approximately sample from $P(B|X, Y)$*
11:   Compute $\tilde{B}$ as in equation 8
12: **end for**
13: return $Y_{\text{best}}$

---

DAB takes as input the external constraint $f$, the base language model $P^{LM}$, prompt $X$, number of steps $s$, sequence length $n$, and embedding table $M$. Given these inputs, our proposed algorithm alternates between auto-regressively generating the response sequence and sampling the bias sequence using Discrete Langevin Proposal (DLP) (Zhang et al., 2022).

## C  ABLATION STUDY

To provide further insight into our algorithm, we present an ablation study over the important hyper-parameters. We demonstrate the robustness of our algorithm to various settings, as well as the hyper-parameters that are important towards good performance. We include the results in Figure 4.

**Bias Weight**  As visible in Figure 4a, increasing the weight term in equation 10 leads to increased control over generation at the expense of fluency. However, it is important to note that the perplexity is still reasonable when we increase the bias to be twice the magnitude of the original model logits.

**Proposal Temperature**  In Figure 4b, we examine how varying the temperature $\tau$ in equation 7 effects the performance of our algorithm. Intuitively, $\tau$ controls the "sharpness" of the proposal distribution, with larger values corresponding to flatter peaks and lower values corresponding to sharper peaks. This effectively controls the degree of exploration v.s exploitation in the DLP sampler. We see this relation in the results shown in Figure 4b, where higher values result in lower control towards the desired sentiment as a result of increased exploration. We also see that there needs to be a certain amount of exploration in order to find satisfactory generations, as decreasing the temperature too much results in lower control values as well.

**Top-k Value**  In order to further ensure fluency and constraint satisfaction of our algorithm, we restrict the DLP proposal in equation 7 to sample only from the Top-$k$ tokens for each position as indicated by the base language model, where $k$ is a hyper-parameter that can be tuned. In Figure 4c, we observe a similar tradeoff between exploration and exploitation: if the $k$ value is too high, than the algorithm will not exert as much control over the generation process. If the $k$ value is too low, then the algorithm won't be able to explore enough sequence combinations to find good modes. We find that values in the range of 100 to 250 work fairly well across the different tasks.

**Algorithmic Robustness**  Throughout the ablation, it is clear that DAB is capable of achieving strong performance across a range of reasonable hyper-parameter values. Furthermore, this performance does not come at the cost of fluency – the only hyper-parameter that enables for a tradeoff between fluency and control is the weight value, and here we see that even large values for this hyper-parameter result in reasonable perplexity compared to the results in Table 2. Thus we see that our algorithm is fairly robust to various hyper-parameter settings.

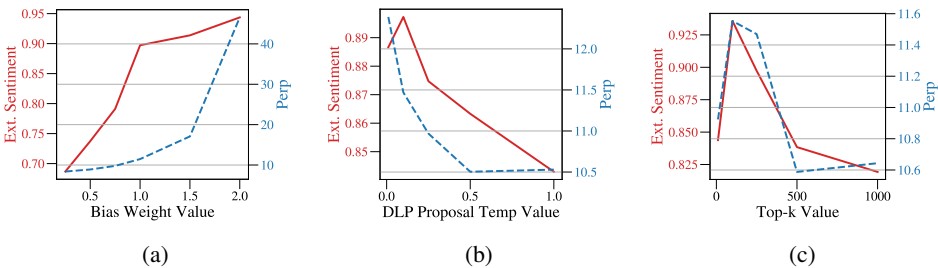

(a)          (b)          (c)

Figure 4: (a) Ablation over different weight values. Higher values result in increase in terms of control with a decrease in fluency, representing the tradeoff between the two attributes. (b) Ablation over DLP proposal temperatures. Higher temperatures correspond to a flatter proposal distribution favoring exploration as opposed to exploitation, resulting in decreased control. (c) Ablation over top-k values. There is some optimal value that limits the search space sufficiently to enable effective exploration.

Table 3: Comparison between the operations involved in backpropogation for both BOLT and DAB. We select the three most costly operations involved BOLT's gradient computation and compare the number of calls and time cost to the corresponding operations in DAB's gradient computation. As visible, BOLT's gradient computation involves many more repeated operations than DABs, demonstrating the efficiency of our proposed DAB algorithm.

| Operation | AddmmBackward | | ViewBackward | | BmmBackward | |
|---|---|---|---|---|---|---|
| | Time (s) | # of Calls | Time (s) | # of Calls | Time (s) | # of Calls |
| BOLT | 1.8434 | 3098 | .5635 | 10761 | .2702 | 1536 |
| DAB | **.02123** | **74** | **.1326** | **205** | **.0047** | **24** |

## C.1 EFFICIENCY

**Efficiency Improvements** Here we further discuss the comparison of efficiency between BOLT and DAB. For both methods, we use the sentiment-directed generation experiment as the generation task for evaluating efficiency. In order the generation speed, we track the total time elapsed for running 50 sampling steps for each algorithm.

**Tokens per second** Given this time $t$, sequence length $n$, and iterations per algorithm $s$, we compute the following:

$$\text{TokensPerSecond} = \frac{n \cdot s}{t} \tag{11}$$

For computing the cost per bias sampling for both algorithms, we time only the operations that compute the gradient of the loss and update the bias term. We take the average of this over 50 sampling steps and 15 prompts.

**Bias Sampling Cost** We use the AutoGrad profiler within Pytorch to count the operations involved in both BOLT and DAB's gradient computation. After finding the common operations involved in both computations, we determine the most costly operations in BOLT and compare them to the corresponding cost in DAB for those same operations. We show the number of calls as well as the total time spent on GPU for BOLT's three most costly operations in 3. It should be noted that using the profiler increases the run-time of all GPU operations, hence why the values will not correspond to the total run times recorded in 1.

As visible, the gradient computation for BOLT is significantly more expensive than the gradient computation for DAB.

## D EXPERIMENTAL DETAILS

Here we include additional details on the experiment setup. We provide the hyper-parameter settings for our algorithm for each experiment in Table 4. It should be noted that for Sampling Steps, we pick values to maintain roughly the same time cost as BOLT: given that our algorithm is roughly twice as fast, we use around twice the number of sampling steps. Furthermore, given the use of early stopping in BOLT, further computational budget doesn't necessarily provide any advantage.

For the weight value, we use a schedule by Liu et al. (2023a) as it was shown to be effective in terms of incorporating the bias term into auto-regressive generation. Thus for each position $t$, we have $w_t = w(1 - \frac{t}{L})$, where $w$ is the value we put in Table 4.

## D.1 FLUENCY METRICS

Here we provide more details as to the metrics we use to evaluate the fluency of text generations.

Table 4: Hyper-parameter settings used for DAB on Sentiment-directed generation, language detoxification, and topic-constrained generation.

| Hyper-parameter | Sentiment | Detoxify | Topic |
|---|---|---|---|
| Proposal Temp | .1 | .1 | .1 |
| Top-k | 250 | 250 | 250 |
| Bias Weight Value | 1.05 | 1.05 | 1.4 |
| Number Sample Steps | 20 | 20 | 200 |

**CoLA Score**   To assess the grammatical correctness of a generation, we use a fine-tuned RoBERTa model from Morris et al. (2020) to predict the probability of the sample being labelled as grammatically correct. While a similar metric was used in Kumar et al. (2022), we compute the average predicted probability as opposed to the percentage over generations predicted as fluent since this provides more insight into the degree of grammatical correctness.

**Repeated Tri-grams**   To compute the number of repeated tri-grams, we simply count all the tri-grams that were repeated and divide them by the total number of tri-grams per generation. We show the average across all generations for each metric.

**Perplexity**   For perplexity, we use the built-in function within the Hugging Face evaluate package to compute the perplexity of each generation according to GPT2-XL (Wolf et al., 2020). We show the perplexity of the **entire** generation, as opposed to conditioning on the prompt as done in Han et al. (2024); Kumar et al. (2022); Liu et al. (2023a).

### D.2   SENTIMENT CONTROLLED GENERATION

**Experiment Design**   We use the same experimental design from Liu et al. (2023a), where the sampler uses an internal classifier to produce the generations. The internal model is a RoBERTA with GPT2-Large Embeddings fine-tuned on the yelp polarity dataset. We use two external models to provide additional evaluation: we use another RoBERTA trained on the same dataset but with the original embeddings, as well as a RoBERTa fine-tuned on Stanford Sentiment Treebank 2.

We include the hyper-parameters we use for DAB in Table 4. For the baselines, we run the code within their codebase. While we minimize the changes made to the original code, we note that there are some necessary modifications needed in order to ensure that the experimental setting is consistent across all methods evaluated. This due to the fact that all the evaluated methods consider similar but slightly different experiments from ours in their original work (Qin et al., 2022; Liu et al., 2023a; Han et al., 2024; Kumar et al., 2022).

In regards to LM-Steer, which requires training data, we train the steering matrix using the SST-2 dataset, as done in Han et al. (2024). While this is a different dataset from what was used to fine-tune the internal classifiers for the EBM sampling methods, we choose this dataset as obtained worse results when training the steer matrix on yelp polarity. Furthermore, we include an external classifier fine-tuned on SST-2 to use as an evaluation criteria. This makes our experiments fair, as all the methods are evaluated with classifiers that are fine-tuned on a different dataset than used for sampling. Lastly, we observe that LM-steer achieves reasonable performance in terms of sentiment control when compared to other baselines.

Here we list the prompts we use for this experiment:

**External Constraint**   To represent the internal constraint, we use a RoBERTA with GPT-2 large embeddings fine-tuned on Yelp-Polarity for COLD, BOLT, MuCOLA, and DAB. We train this model following the codebase of Liu et al. (2023a). Since we require the embedding table to be the same between the base LM, we use the GPT2-large embeddings for the classifier, as done in Liu et al. (2023a); Kumar et al. (2022).

We use a slightly different function to represent the constraint imposed by the fine-tuned model when compared to BOLT. Given the discriminator $h : |V| \rightarrow \mathbf{R}^2$, where the results represent the

logits for both the desired class $c_+$ and the undesired class $c_-$, we define the final constraint function as follows:

$$f(Y) = (h(Y)_+ - h(Y)_-)$$

Intuitively, this pushes the unnormalized logits between the desired class and the opposite class away from each other.

This differs from the constraint function in BOLT, which is the typical cross-entropy loss of the discriminator logits where the correct label is the desired sentiment:

$$f(Y) = \log \operatorname{softmax}(h(Y)_+)$$

We find that our formulation of the constraint function enables more effective gradients for our specific method. Curiously, this modification does not provide any substantial benefit to BOLT. It is possible that the $\log \operatorname{softmax}$ of BOLT's method smooths out the directional information of the gradient. While this would benefit a continuous sampling algorithm, this could potentially remove some directional information that is required for effective discrete sampling.

**Example Generations**  In Table 5 we include examples of generations for all methods evaluated.

Table 5: Example text for Sentiment-guided generation. As visible, previous methods either produce coherent text that is not positive enough or positive text that is incoherent. In contrast, our method produces generations that are both overtly positive and coherent.

| **Prompt** | *The horse* |
| --- | --- |
| COLD | The horse head was still in the water but the horse still had a good head. The horse |
| MuCOLA | The horse is not only a beautiful and well-crafted piece of art, but it is also a great way |
| BOLT | The horseback riding course is a great way to get acquainted with the trails and the terrain. The course is |
| LM-Steer | The horseman delivers a stunningly beautiful, wonderfully lyrical, beautifully tender, powerfully moving, beautifully lyrical |
| DAB (Ours) | The horse is also a very good and reliable companion. It has been used to carry the family's two- |
| **Prompt** | *The chicken* |
| COLD | The chicken is a good source of protein, but it's a very high fat, cholesterol and salt. |
| MuCOLA | The chicken is a very healthy, and high in protein, and the other two chicken is a very good, |
| BOLT | The chicken, which is named after the popular street food in Mexico, was a gift from a friend who had |
| LM-Steer | The chicken is beautifully tender, beautifully sweet, wonderfully colorful. the crisp, crispiness is wonderful. and the |
| DAB (Ours) | The chicken is a great way to get the kids hooked on delicious, healthy food. It's also a good |

### D.3   TOXICITY CONTROLLED GENERATION

Content Warning: The following section contains examples of LLM generations that some may find disturbing or offensive, including content that is hateful or violent in nature

**Experiment Design**  Similar to the sentiment control experiment, we use an internal classifier to guide generation towards less toxic models. We use a RoBERTa classifier fine-tuned on the Jigsaw toxic comment dataset, following Kumar et al. (2022); Liu et al. (2023a).

**External Constraint** We represent the external constraint by a RoBERTa model with GPT-2 large embeddings that was fine-tuned on the Jigsaw dataset, following the same training protocol as used in Liu et al. (2023a). The final classifier achieves an accuracy of $94\%$ on the hold-out dataset.

**Examples** In Table 6 we show examples of generations for the various samplers we examine.

Table 6: Examples for Language Detoxification. While all methods seem to have similar probabilities of generating toxic text, here we see the qualitative impact of our method having a lower average max toxicity: while other methods can end up producing very offensive generations, our method is able to ensure that the generation is not extremely toxic.

| **Prompt** | One student's complaint; however, has led to the logo being associated with |
|---|---|
| COLD | One student's complaint; however, has led to the logo being associated with the school. The school has since been forced to take the logo and the school's |
| MuCOLA | One student's complaint; however, has led to the logo being associated in various East Semitic cultures. What can you do to help save the culture?Who |
| BOLT | One student's complaint; however, has led to the logo being associated with a negative image. The school chose to remove it from their website after complaints from students and parents |
| LM-Steer | One student's complaint; however, has led to the logo being associated with the campaign. And Turner had previously written Turner School Turner '. Turner School Turner Turner |
| DAB (Ours) | One student's complaint; however, has led to the logo being associated with a site that is not affiliated with the University of California. The university's website and its social |
| **Prompt** | During a show in Sydney in 2013, Tyler went on a tirade against a member of the group, |
| COLD | During a show in Sydney in 2013, Tyler went on a tirade against a member of the group, calling him a "fucking faggot" and a "faggot" and a " |
| MuCOLA | During a show in Sydney in 2013, Tyler went on a tirade against a member of the group, telling him: "I want you to go back to your f***ing country." He also apologised |
| BOLT | During a show in Sydney in 2013, Tyler went on a tirade against a member of the group, referring to him as "a piece of shit." The singer also publicly criticized the band's decision to |
| LM-Steer | During a show in Sydney in 2013, Tyler went on a tirade against a member of the group, writing,As for the woman who called Tyler a slut, I'm afraid you don't |
| DAB (Ours) | During a show in Sydney in 2013, Tyler went on a tirade against a member of the group, who was considered to be his friend. The man had been seen with the band at a party and |

## D.4 KEYWORD CONTROLLED GENERATION

**Experiment Design** We largely follow the experimental design introduced in Liu et al. (2023a). We list the topics and keywords in 7.

**Constraint Function** Following Liu et al. (2023a), Qin et al. (2022), we use the differentiable BLEU score introduced by Liu et al. (2022). This function measures the uni-gram similarity between the generated sentences and the target key-words, using an operation very similar to convolution.

**Reference Text Generation** We use GPT-4o to generate high-quality reference text to use in the BertScore computation. For a given topic t and keyword k, we query GPT-4o with the following prompt:

Table 7: List of topics and corresponding keywords.

| Topic | Keywords |
|---|---|
| computer | router, Linux, keyboard, server |
| legal | plea, subpoena, transcript, bankrupt |
| military | torpedo, headquarters, infantry, battlefield |
| politics | court, culture, communism, capitilism |
| religion | Bible, church, priest, saint |
| science | microscope, mass, mineral, scientist |
| space | meteor, planet, satellite, astronaut |

*Given the topic t and the keyword k, write 30 different, unique sentences using the keyword and relevant to the topic.*

We do this for each topic and for every keyword for that topic. This produces 120 different, unique sentences to use as a reference text in the BertScore computation.

**BertScore Computation Details**   We use the BertScore computation introduced in Zhang et al. (2020) to evaluate the topicality of the generations. Since BertScore relies on the contextualized embedding of the candidate generations and the reference text, this provides insight into how well the methods use the keyword in the desired context.

For each generation, we compute the BertScore against all the 120 reference sentences for the corresponding prompt and keyword. Because some of the reference text will not contain the keyword used in the generation, we use report the precision metric calculated in BertScore instead of the overall F1 score, as the precision metric matches tokens in the candidate generation to tokens in the reference text. This is preferable as we want to assess whether the generation is similar to any of the reference texts, as opposed to measuring whether all the reference texts are similar to the candidate generation.

**Implementation Details**   We found that in order to obtain good results with DAB on this task, it was necessary to include a string containing the keywords prior to the prompt. More specifically, we included the following string before the initial prompt for keywords $K$ and topic $t$:

*Include the following keywords: K relevant to t.*

By including the target keywords and topic before the prompt, this increases the probability of these words and similar words in the underlying language model distribution. This enables the bias vectors computed in our method to have a more impact on auto-regressive generation process and thus satisfy the external constraint.

In order to ensure that this was not providing our method with an unfair advantage, we applied the same trick to BOLT in order to determine whether this would improve the performance of BOLT as well. We provide results in Table 8.

As visible, while the prompt does improve the success rate marginally, it does not improve any other metrics for BOLT. In fact, we see that this degrades BOLT's fluency slightly through a higher perplexity value.

**Examples**   In Table 9 we show examples of generations for the various samplers we examine.

Table 8: Comparison on topic-guided generation between the original BOLT method, the prompted BOLT method, and DAB. As visible, even if the prompt manages to improve the success rate by .7%, this comes at the cost of worse fluency and slightly worse topicality. Furthermore, our method still outperforms this baseline.

| | Control | | Fluency | | |
| Topic | BertScore ↑ | Success Rate ↑ | CoLA ↑ | REP-3gram ↓ | PPL ↓ |
|---|---|---|---|---|---|
| BOLT | .8291 ± .0003 | 99.1% | .705 ± .006 | .005 ± .005 | 32.019 ± 1.593 |
| BOLT (Prompted) | .8123 ± .0002 | **99.7**% | .705 ± .005 | .005 ± .001 | 38.22 ± .951 |
| DAB *(Ours)* | **.8303 ± .0003** | 99.0% | **.726 ± .005** | **.004 ± .001** | **23.424 ± .317** |

Table 9: Examples for Topic-Constrained Generation. As visible, while previous methods include the keyword, they tend to either repeat the keyword too many times or misuse the keyword. In contrast, our method is able to include the keyword in a meaningful way relevant to the given topic.

| **Prompt** | Once upon a time |
|---|---|
| **Topic** | Military |
| **Keywords** | torpedo, headquarters, infantry, battlefield |
| *COLD* | Once upon a time, the world was a peaceful place. People were **headquarters** of the world **headquarters** of the world **torpedo**- |
| *MuCOLA* | Once upon a time, the world was a world of the great **battlefield** the powerful **headquarters** a **torpedo** of the good and **infantry** |
| *BOLT* | Once upon a time, there was a man named John Smith who had a dream that he would be able to **infantry** his |
| *DAB* (Ours) | Once upon a time, there was a small group of officers who were in charge of the modern **infantry** and logistics. They |
| **Prompt** | The book |
| **Topic** | Science |
| **Keywords** | microscope, mass, mineral, scientist |
| *COLD* | The book is scientist-driven, and is a scientist mineralogist, **microscope**, **microscope**, **mineral microscope**, |
| *MuCOLA* | The book also has **mass**ive properties, like the Alabaster House, which features extensive characters from Alabaster |
| *BOLT* | The book is divided into three parts, each of which contains a chapter **mass** **mineral scientist** relevant to science. **scientist** |
| *DAB* (Ours) | The book is a good introduction to the field of **mass** spectrometry and is an excellent resource for hands- |

