# OpenReview forum: "Controlled LLM Decoding via Discrete Auto-regressive Biasing"
_ICLR.cc/2025/Conference — ICLR 2025 Poster_

### Official Review · Reviewer_RyTz · 2024-11-03

**Soundness:** 4
**Presentation:** 4
**Contribution:** 4
**Rating:** 8
**Confidence:** 4

**Summary:**

The paper presents an algorithm for constrained autoregressive decoding where the constraint is defined by a distribution function. This is a common approach using energy-based models.

Even though BOLT does propose similar algorithm, the main difference in this paper is that instead of sampling the bias in the continuous domain, the bias token sampling in this paper happens in discrete space. Author claim, and substantiate with experiments, that this produces sequences that not only follow the constraint better, but also the LLM policy model leading to more fluent outputs.

**Strengths:**

The authors propose DAB - Discrete Autoregressive Biasing, a modification of BOLT where both output sequence Y and the biasing B are always sampled in discrete space. This allows the model to always remain in the discrete space leading to not just lower computational cost compared to other energy based decoding methods, but also improve the generation with respect to fluency and constrain.

As part of DAB:
1. A joint distribution over the output Y and Bias tokens B is proposed.
2. Similar to Gibbs sampling, the proposed sampling algorithm (Langevin within Gibbs) alternates between sampling better Y and B while at the same time using Langevin for prediction the distribution to sample the tokens from.
3. Uses MCMC to sample the bias tokens, and then add bias from those while sampling the response token.

Albations are performed on both hard and soft constrains to show the effectiveness of the model.

**Weaknesses:**

- Even though the method is fast compared to other energy based models, the method is still slow because of the autoregresive sampling happening inside the step for loop as shows in Algorithm 1, line 3-7.
- Results on Sentiment are mixed with decreased Fluency compared to BOLT.

**Questions:**

In equation 8, the bias value b_{i,j} corresponding to any bias token b_{i} is calculated as the L2 distance between their corresponding embeddings. LLM loss functions use inner product between the embeddings as the distance metric while calculating logits. Any reason to use L2, and were other options tried?

---

> ### Author Response · Authors · 2024-11-21
> **Response**
>
> Thank you for your supportive and thoughtful review. We respond to your points below:
>
> **Weakness: Auto-regressive sampling**
>
> This is correct — the main deficiency of our method is the fact that each step requires the autoregressive generation of the sequence. Prior work also suffers from this issue [1]. Improving efficiency by reducing the number of auto-regressive calls is an interesting future direction.
>
> **Weakness: Mixed results on Sentiment Task**
>
> We would like to point out that while our method underperforms BOLT slightly in the context of fluency, our method greatly outperforms BOLT in terms of sentiment control while almost matching BOLT’s fluency performance. While our method has the second-best fluency metrics, BOLT lags behind other baselines for most of the sentiment-specific metrics. Additionally, as shown in the qualitative examples we include in Table 6 of the Appendix, DAB generations are not noticeably less fluent than BOLT generations. Finally, DAB outperforms BOLT in terms of fluency on the keyword generation task, as shown in Table 2. We believe that this demonstrates that our method is able to achieve a superior balance between control and fluency.
>
> **Question: Alternative Calculation of Bias Vector**
>
> We chose to use the $l_2$ distance as this created a much more peaked distribution towards the desired token. When we tried using the inner product, it did not enable sufficient control as the bias vector did not bias strongly enough towards the sampled tokens. We will add this result to the appendix.
>
> [1] Liu et al. BOLT: Fast Energy-based Controlled Text Generation with Tunable Biases. ACL 2023.

---

### Official Review · Reviewer_HCm1 · 2024-11-04

**Soundness:** 2
**Presentation:** 1
**Contribution:** 2
**Rating:** 3
**Confidence:** 3

**Summary:**

This paper proposes a new controlled text generation approach called DAB. DAB happens at decoding time, and it consists of two alternating steps: updating the bias term based on the gradient, and updating the response sequence conditioned on the bias term. This approach is designed to address the trade-off between fluency and control satisfaction. They experiment on sentiment, toxicity, and lexical control benchmarks.

**Strengths:**

- good summary of prior work, it nicely summarizes the field of controllable text generation from AR to NAR, with or without gradient guidance, etc.
- the method makes intuitive sense, but the details of the methods seem very unclear.

**Weaknesses:**

- Controllable text generation is important, but the benchmarks this paper tested have been mostly saturated by prior approaches, so further testing on these benchmarks can no longer demonstrate the goodness of the newly proposed approach. Furthermore, prompting could solve these problems, so can this approach solve even harder problems, like model jailbreaking via such MCMC type of approach?

- I think the math is not very solid in the paper. equation 6 seems wrong. It's a circular definition. The equation (not numbered) immediately after 6 is also strange, should it be P(B | Y, X)? There are also unclear notations in the method section.

- The method section is very badly written. I don't understand many technical details in the method section. Why eqn 7 has the (1-bij) term? Why is eqn 10 argmax for a sampling distribution? If B is the bias term, intuitively initializing B at 0 seems more reasonable than initializing B at Y?

**Questions:**

see the weakness section.

---

> ### Author Response · Authors · 2024-11-21
> **Response to Saturated Benchmarks, Notational Errors**
>
> Thank you for your helpful comments. We would like to address some of your concerns to clarify the technical details and mathematical formulation.
>
> **Saturated benchmarks**
>
> We are unclear about your statement regarding benchmarks being "saturated." Do you mean there are too many methods addressing the same task? We believe the tasks we selected are appropriate as they allow for a clear comparison of the difference in performance between controlled text generation algorithms. As shown in Table 2, these tasks are useful for understanding where prior methods fall short and where our method excels. Furthermore, the use of similar tasks in prior works demonstrates that our chosen benchmarks are reasonable [1, 2, 5, 6].
>
> Furthermore, regarding your claim that prompting can solve these tasks, we are not aware of any papers that demonstrate prompting is sufficient for the tasks we consider. In fact, [1] does compare to Prompt-T5, a pre-trained LM intended to solve tasks with prompts. They demonstrate that LM-Steer outperforms Prompt-T5, which shows that prompting is not enough to match controlled text generation methods. We choose not to compare to Prompt-T5 as [1] already establishes the limitations of prompt-based methods.
>
> Finally, model jailbreaking is a separate research direction from the core focus of this paper. None of the methods we compare against include jailbreaking as a task [1, 2, 5, 6]. We consider it an application of our proposed method that could be investigated for future directions.
>
> [1] Han et al. Word Embeddings Are Steers for Language Models. ACL 2024.
>
> [2] Liu et al. BOLT: Fast Energy-based Controlled Text Generation with Tunable Biases. ACL 2023.
>
> [3] Stidikov et al. Classifiers are Better Experts for Controllable Text Generation. Workshop on Transfer Learning for NLP, 2022.
>
> [4] Dekoninck et al. Controlled Text Generation via Language Model Arithmetic. ICLR 2024.
>
> [5] Kumar et al. Gradient-based Constrained Sampling from Language Models. ACL 2022.
>
> [6] Qin et al. COLD Decoding: Energy-based Constrained Text Generation with Langevin Dynamics. NeurIPS 2022.
>
> **Notational errors**
>
> Thank you for the feedback. While equation 6 is correct, we understand that it can be confusing as it is not very direct. We will change it to “$P(B|X)$ is defined to be $ \frac{\exp(f(B | X))}{Z_B}$”. The equation that immediately follows should be $P(B | X, Y)$ as you have pointed out. We occasionally drop the conditioning on X as all terms should be understood as conditioned on prompt X. We have revised the notations in the paper to make them clearer.

---

> ### Author Response · Authors · 2024-11-21
> **Clarification of Method Section**
>
> **Discrete Langevin Proposal function**
>
> The $(1 - \hat{b}\_{ij})$ term corresponds to the hamming distance between tokens. We include this term as a result of the definition of Discrete Langevin Proposal (DLP) introduced by [1]. For your convenience, we will explain how we obtain the proposal function below.
>
> To enable the use of large step sizes in the proposal, we adopt the globally balanced version of the DLP proposal [1, 2]:
>
> $
> \text{Categorical} \left( \underset{j \in |V|}{\text{softmax}} \left( \nabla f(\hat{B} | X)_i (\text{Onehot}_j - \hat{b}_i)\right) \right)
> $
>
> Here, $\text{Onehot}_j$ represents the one-hot vector for the $j$ token in the vocabulary $V$, and $\hat{B} = \{\hat{b}_1,\hat{b}_2, … \hat{b}_n \}$ represents the original one-hot vector sequence of length $n$. In order to obtain a distribution over the vocabulary $|V|$, we must compute the inner term for every token $j \in V$.
>
> We note that $(\text{Onehot}_j - \hat{b}_i))$ corresponds to the distance between the original token at position $i$ and every token $j$ in the vocabulary $V$. Given the discrete nature of the tokens, we choose to use hamming distance to represent this term. For a token $j$, the hamming distance to the original token in position $i$ is 0 if the $j$th coordinate $\hat{b}\_{ij} = 1$, as they are the same token; and 1 if the $j$th coordinate is 0. Thus we can represent the hamming distance between token $j$ and the current token as $1 - \hat{b}\_{ij}$. Below we include the proposal distribution we sample from to obtain the new token for position $i$.
>
>
> $
> b’_i \sim \text{categorical} \left(\underset{j \in |V|}{\text{softmax}} \left( \frac{1}{\tau} (\nabla f(\hat{B} | X))\_{ij} (1 - \hat{b}\_{ij}) \right) \right)
> $
>
> Here, $b’_i$ is the token we sample from the categorical distribution over V on the right-hand side.
>
> **Use of greedy decoding**
>
> Equation 10 is an argmax to represent greedy decoding, which is a commonly used decoding technique for auto-regressive generation. While other sampling methods are compatible with our algorithm, we choose greedy decoding as previous inference-time controlled generation algorithms [3, 4], also use greedy decoding. This helps emphasize the novel aspects of our framework and ensures a fair comparison with previous works.
>
>
> **Initialization of $B$**
>
> To answer your question regarding why we initialize $B$ to $Y$ when sampling from $P(B | X, Y)$, we provide our reasoning in section 4.2 under the section titled “Sampling from $P(B | X, Y)$”. Specifically, our goal is to sample from the distribution:
>
> $
> P(B | X, Y) \propto P^{LM}(Y | X, B) \exp(f(B | X))
> $
>
> As we discuss in the mentioned section, sampling from this distribution would require computing $P(Y | X, B)$ for all possible values of $B$, which is intractable. In order to obtain a more feasible calculation, we note that $P^{LM} (Y | X, B)$ will be high when the bias $B$ aligns with the original response $Y$ due to the nature of auto-regressive generation. Thus we approximate this distribution by initializing $B = Y$, which will ensure a relatively high value for $P^{LM} (Y | X, B)$. By initializing the bias term into a region with high values of $P(Y | X, B)$, all that remains is to determine which samples within this region enable better constraint satisfaction. If we initialize $B=0$, then sampling must simultaneously find $B$ that results in high $P(Y | X, B)$ and high  $f(B | X)$, which is a more difficult task.
>
> [1]. Zhang et al. A Langevin-like Proposal for Discrete Spaces. ICML 2022.
>
> [2]. Pynadath et al. Gradient-based Discrete Sampling via Automatic Cyclical Scheduling. NeurIPS 2024.
>
> [3]. Liu et al. BOLT: Fast Energy-based Controlled Text Generation with Tunable Biases. ACL 2023.
>
> [4]. Qin et al. COLD Decoding: Energy-based Constrained Text Generation with Langevin Dynamics. NeurIPS 2022.
>
> We appreciate the time taken to point out the notational errors in our work and areas of confusion. We were wondering if there were any other flaws that lead to your score of a 3 — are there any serious concerns you have with the method itself? Were there any concerns with the claims made in our paper and whether we provide sufficient evidence to validate them? We would greatly appreciate your input in helping us improve our work.

---

### Official Review · Reviewer_w6Rd · 2024-11-04

**Soundness:** 2
**Presentation:** 2
**Contribution:** 2
**Rating:** 6
**Confidence:** 3

**Summary:**

The paper proposes an approach to controlled text generation --- DAB (Discrete Auto-regressive Biasing) --- that exploits the DLP (Discrete Langevin Proposal) technique from (Zhang et al. 2022) for efficiently sampling inside a discrete space while still being able to exploit gradients of an energy function over this space. The DLP technique is not used directly over the output sequence, but over an auxiliary "bias sequence" that is coordinated with the output sequence through a Gibbs-Sampling-like alternation. The experiments show competitive results with a few baselines in terms of efficiency as well as balance between constraints and fluency of the obtained results.

**Strengths:**

The main strength of the paper lies in its innovative application of the DLB technique to the general problem of controlled text generation (Zhang et al. did touch on text generation but only in a very limited way).

The paper is also creative in the way it uses an auxiliary bias sequence to steer the generation of the actual output sequence.

The experiments demonstrate certain advantages in terms of control (i.e. constraint satisfaction) and fluency over a number of baselines, in particular some based on gradient techniques over continuous relaxations of EBMs over discrete spaces.

**Weaknesses:**

The main weaknesses of the paper lie on two dimensions: (A) omission of significant related work, (B) lack of discussion/clarity about certain key aspects and modelling decisions in the paper.

(A) The Related Work section 2.1 devoted to Language Models as EBMs, totally ignores a substantial line of work specifically devoted to *discrete* sampling from EBMs, either (i) with focus on training autoregressive approximations to these EBMs (exemplified by [1] and a number of more recent publications at ML conferences (see references in [3])), or (ii) with focus on decoding-time techniques [2]. This line of work, like the present paper, is concerned with discrete (as opposed to continuous) sampling, and is not limited to encoder-based architectures.

[1] Khalifa et al. A distributional approach to controlled text generation. ICLR 2021.

[2] Eikema et al. An approximate sampler for energy-based models with divergence diagnostics. TMLR 2022.

[3] Kruszewski et al. disco: a toolkit for Distributional Control of Generative Models. ACL 2023.


(B.1) Concerning the core Equation (5). First, in the expression $P^{LM}(Y|X,B)$, what you seem to mean is that you concatenate the input $X$ with the bias sequence $B$ and then apply the LM on this new input, but it would be worth discussing this assumption. Second, and more importantly, while it is not clear at this point in the paper, it seems that in Algorithm 1 you actually need to compute $f(Y)$, and not only $f(B)$. Then a pretty obvious question for the reader is, why not simply define $P(Y|X) \propto P^{LM}(Y|X) \exp(f(Y))$ ? That would be much more direct than what the paper does, would directly define $P(Y|X)$ as an EBM, and then presumably the DLB technique could be directly applied to this EBM. It is not clear to me why the authors do not consider and discuss this possibility.
(Of course, several other techniques for sampling from this EBM would be possible, including those mentioned in (A) and techniques related to RLHF/PPO where $f(Y)$ might be seen as a reward.)

(B.2) There are several other points in the paper that are kept implicit and would need more discussion. To give a few examples:
- The fact that the length $n$ of the response $Y$ needs to be specified in advance, which appears to be a limitation of the approach, is kept implicit.
- The important DLB-based equation (7) should be described in a more self-contained way (perhaps using the Appendix), in particular the way the gradient is actually computed.
- The fact that step 6 in Algorithm 1 is actually deterministic should be mentioned, as this detracts from standard Gibbs-sampling practice.

**Questions:**

Questions/suggestions (in addition to those implicit in the previous section):

- Lines 238-243 seem problematic, as they introduce a notation $P(Y|B)$ that is not conditioned by $X$. Are they correct and/or needed later?

- In Lines 494-496, you mention that a good metrics for the keyword-guided generation should consider the meaning similarity of the produced sentence to the constraining keywords, not the actual presence of the keywords. I was not fully convinced by this remark, and was wondering whether considering the actual presence of the keywords should be seen as a more important metrics. More generally, I wondered whether it would be worth reporting, among the metrics, the value $f(Y)$ itself as this seems to be the main driver of the approach.

---

> ### Author Response · Authors · 2024-11-21
> **Response to A: Related Works**
>
> Thank you for providing [1], [2], and [3]. While we did discuss works that fine-tune the model very briefly, we were unaware of these specific works and will include them in the revision, as they add more breadth to our discussion on controlled text generation methods.
>
> We want to emphasize that while the suggested papers are related, they are not closely aligned with the focus of our work. As you pointed out in your review, [1], [3] focus on methods that fine-tune auto-regressive models to align with some defined EBM. Our work is primarily concerned with inference-time decoding algorithms that do not require the model to be fine-tuned.
>
> We would also like to point out that the algorithm presented in [2] requires some proposal function that can be used to generate samples. Its core contribution is an accept / reject algorithm that is agnostic to the proposal function. This is different from our algorithm, which presents a novel decoding algorithm that directly generates new samples. Thus we compare primarily to other works that introduce decoding methods that generate samples, which is not the focus of [2]. Nevertheless, we will make sure to include all three works in our related work section as they provide valuable insight into alternative approaches to this problem.
>
> [1] Khalifa et al. A distributional approach to controlled text generation. ICLR 2021.
>
> [2] Eikema et al. An approximate sampler for energy-based models with divergence diagnostics. TMLR 2022.
>
> [3] Kruszewski et al. disco: a toolkit for Distributional Control of Generative Models. ACL 2023.

---

> > ### Author Response · Authors · 2024-11-21
> > **Response to Questions**
> >
> > **Q1: Notation**
> >
> > Thank you for pointing this out. The correct term should be $P(Y | X, B)$. We occasionally drop the conditioning on $X$ as all terms should be understood as conditioned on prompt $X$ — we have revised the notation in the paper to make it clearer.
> >
> >
> > **Q2.1: Keyword Inclusion**
> >
> > In the mentioned paragraph, we never claim that the semantic similarity to the keyword is more important than the keyword itself. We specifically state “The ideal metric goal for this task should only assign good scores to text where keywords are used in a meaningful way.” Inclusion of the keyword is necessary, but not sufficient — not only should the keywords be included, but it should be included in a meaningful way. Inclusion rate alone is inadequate, as it only captures the inclusion of the keyword, failing to assess whether the inclusion is semantically coherent. Therefore, we use the BERT score to evaluate whether the keywords are included in a meaningful way.
> >
> > **Q2.2: Values of Constraint Function $f(Y)$**
> >
> > For the sentiment and detoxification task, we include the scores assigned to the samples from the internal classifier, or the classifier used to guide the generation process. This directly corresponds to the constraint value for these tasks.
> >
> > Following prior work [1, 2, 3], we do not include the constraint value for the keyword task as it is more difficult to interpret than the success rate and other metrics. The constraint function computes a score based on the probability each logit vector places on the keyword tokens. This value is maximized when all the sequence positions are the keyword token, which is clearly undesirable. As high values and low values can indicate undesirable behavior, this specific constraint function is difficult to interpret. In contrast, the inclusion rate and BERT score are easily interpretable as higher values indicate strictly more desirable behavior.
> >
> > [1] Liu et al. BOLT: Fast Energy-based Controlled Text Generation with Tunable Biases. ACL 2023.
> >
> > [2] Kumar et al. Gradient-based Constrained Sampling from Language Models. ACL 2022.
> >
> > [3] Quin et al. COLD Decoding: Energy-based Constrained Text Generation with Langevin Dynamics. NeurIPS 2022.
> >
> > [4] Liu et al. Don’t Take It Literally: An Edit-Invariant Sequence Loss for Text Generation. ACL 2022.
> >
> > We would like to thank you for your helpful review, and we hope that our response clears up any concerns. If there are any remaining areas of confusion or reasons for concern, we are happy to answer follow-up questions and engage in further discussion.

---

> > > ### Comment · Reviewer_w6Rd · 2024-11-25
> > > **Thank you for your responses.**
> > >
> > > Thank you for your responses and for the paper update. Based on these improvements, I am leaning towards acceptance.

---

> ### Author Response · Authors · 2024-11-21
> **Response to B.1.1: Biased Auto-regressive generation $P(Y | X, B)$**
>
> Our computation of $P(Y | X, B)$ does not involve concatenating the input $X$ with the bias sequence $B$. As we explain in equation 10 in Sec 4.2, we use $B$ as a biasing sequence that is added to the unnormalized logits the language model outputs for each position. Specifically, during the auto-regressive generation, after the model produces the unnormalized logits $\tilde{y_i}$ for position $i$, we add $\tilde{b_i}$ to obtain a new logit vector. We then apply greedy decoding to this logit vector to obtain the token for this position. However, it is possible to apply other forms of sampling — we choose greedy decoding since this is what previous work [1, 2] uses and we want to ensure that comparisons between our method and theirs are as fair as possible.
>
> [1] Liu et al. BOLT: Fast Energy-based Controlled Text Generation with Tunable Biases. ACL 2023.
>
> [2] Quin et al. COLD Decoding: Energy-based Constrained Text Generation with Langevin Dynamics. NeurIPS 2022.

---

> ### Author Response · Authors · 2024-11-21
> **Response to B.1.2: Definition of $P(Y | X)$**
>
> Defining $P(Y | X) \propto P^{LM} (Y | X) \exp f(Y)$ is equivalent to the energy function formulation used in previous works (lines 155-157) with $\lambda_1 = \lambda_2 = 1$.  As you noted, DLP can be used to directly sample from $P(Y | X)$. However, this approach does not support auto-regressive generation and significantly compromises fluency. Non-autoregressive approaches usually suffer from lack of fluency as demonstrated in our experimental results in Table 2 and previous works [1]. Additionally, methods like RLHF or PPO require extra data and fine-tuning, making them unsuitable as inference-time approaches.
>
> The primary motivation behind our framework is the observation that fluency is best satisfied through auto-regressive generation, and gradient-based sampling efficiently finds responses that satisfy constraints. By framing the problem as a joint distribution of $Y$ and $B$, we enable the use of both methods — we use autoregressive generation to obtain $Y$, ensuring fluent generations; and we apply Discrete Langevin Proposal (DLP) to sample $B$, ensuring constraint satisfaction. Furthermore, all of this is accomplished without the need to fine-tune the underlying LM. We will add the above explanation to the revision.
>
>
> [1] Liu et al. BOLT: Fast Energy-based Controlled Text Generation with Tunable Biases. ACL 2023.

---

> ### Author Response · Authors · 2024-11-21
> **B.2: Hidden details**
>
> **Sequence Length:**
>
> We do not emphasize that our algorithm requires a specified sequence length as this is a common requirement within the literature — MuCOLA [1], COLD [2], and BOLT [3] all require the specification of sequence length.
>
> Furthermore, it is possible to enable our algorithm to produce sequences of varying lengths by padding / truncating the bias sequence during generation. We did not focus on enabling dynamic sequence length as this is not a core contribution of our work.
>
> **Discrete Langevin Proposal**
>
> We agree with your advice and have included the following discussion regarding our application of Discrete Langevin Proposal (DLP) in the Appendix in our revision. Specifically, to enable the use of large step sizes in the proposal, we adopt the globally balanced version of the DLP proposal [1, 2]:
>
> $
> \text{Categorical} \left( \underset{j \in |V|}{\text{softmax}} \left( \nabla f(\hat{B} | X)_i (\text{Onehot}_j - \hat{b}_i)\right) \right)
> $
>
> Here, $\text{Onehot}_j$ represents the one-hot vector for the $j$ token in the vocabulary $V$, and $\hat{B} = \{\hat{b}_1,\hat{b}_2, … \hat{b}_n \}$ represents the original one-hot vector sequence of length $n$. To obtain a distribution over the vocabulary $|V|$, we must compute the inner term for every token $j \in V$.
>
> Here, we note that $(\text{Onehot}_j - \hat{b}_i))$ corresponds to the distance between the original token at position $i$ and every token $j$ in the vocabulary $V$. Given the discrete nature of the tokens, we choose to use hamming distance to represent this term. For a token $j$, the hamming distance to the original token in position $i$ is 0 if the $j$th coordinate $\hat{b}\_{ij}= 1$, as they are the same token; and 1 if the $j$th coordinate is 0. Thus we can represent the hamming distance between token $j$ and the current token as $1 - \hat{b}\_{ij}$. Below we include the proposal distribution we sample from to obtain the new token for position $i$.
>
> $
> b’_i \sim \text{categorical}\left(\underset{j \in |V|}{\text{softmax}} \left( \frac{1}{\tau} (\nabla f(\hat{B} | X))\_{ij} (1 - \hat{b}\_{ij}) \right) \right)
> $
>
> Here, $b’_i$ is the token we sample from the categorical distribution over V on the right-hand side. We have incorporated this discussion into the appendix, following your advice.
>
> **Gradient Computation**
>
> Once we represent the sequence of tokens $B$ as a sequence of one-hot vectors $\hat{B}$, we are able to compute the gradient of the constraint function with respect to $\hat{B}$ using automatic differentiation software, such as Torch Autograd.
>
> **Use of Greedy Decoding**
>
> We define the biased auto-regressive generation used in our algorithm in terms of an argmax as shown in equation 10, which should convey that this specific component of our algorithm is deterministic.
>
> Additionally, it should be noted that while we choose greedy decoding, our method is compatible with other decoding approaches that are conventionally used, such as top-k, top-p, and standard Gibbs. We choose to use greedy decoding since this was used by prior work [2, 3] and it places emphasis on the novel aspects of our framework.
>
> [1] Kumar et al. Gradient-based Constrained Sampling from Language Models. ACL 2022.
>
> [2] Quin et al. COLD Decoding: Energy-based Constrained Text Generation with Langevin Dynamics. NeurIPS 2022.
>
> [3] Liu et al. BOLT: Fast Energy-based Controlled Text Generation with Tunable Biases. ACL 2023.
>
> [4] Zhang et al. A Langevin-like Sampler for Discrete Spaces. ICML 2022.
>
> [5] Pynadath et al. Gradient-based Discrete Sampling by Automatic Cyclical Scheduling. NeurIPS 2024.

---

### Official Review · Reviewer_3EGw · 2024-11-05

**Soundness:** 2
**Presentation:** 3
**Contribution:** 3
**Rating:** 6
**Confidence:** 2

**Summary:**

This paper introduces DAB (Discrete Auto-regressive Biasing), an algorithm for controlled text generation with large language models (LLMs).  Previous methods often use energy-based decoding in continuous space, which struggles to balance fluency and constraint satisfaction. DAB addresses this by operating entirely in the discrete space of text tokens.

DAB samples from the joint distribution of generated sequence Y and an auxiliary bias sequence B and alternate between biased auto-regressive generation and discrete gradient-based sampling. Specifically, given a generated text Y, the gradient-based discrete sampling is used to maximise constraint satisfaction. Then B is fixed, and biased auto-regressive generation is used to sample Y. A penalization is applied based on sampled tokens' distance from B in embedding space.

Experiments show DAB outperforms baselines such as BOLT and LM-Steer on sentiment-controlled generation, language detoxification, and keyword-guided generation.

**Strengths:**

* The paper identifies previous work's deficiency in balancing fluency and constraint satisfaction and proposed a method that maximises both (according to several benchmark numbers).

* DAB seems to be more stable and robust than other continuous methods.

**Weaknesses:**

Nothing major as I can see.

**Questions:**

N/A

---

> ### Author Response · Authors · 2024-11-21
> **Response**
>
> Thank you for your supportive review. We were wondering if there were any aspects of our paper that prevented you from giving it a higher score. We would be more than happy to address any questions or concerns you may have regarding our paper.

---

### Author Response · Authors · 2024-11-21
**Summary Response**

We would like to thank the reviewers for their constructive responses. We have provided the revised version of our submission, with modifications highlighted in blue. Below we summarize the primary modifications:

1. The related works section is revised to more thoroughly discuss the field of controlled text generation.
2. The equations in Section 4 are revised to be more notationally consistent and informative
3. Section 4 is revised to include additional motivation for our formulation of the target distribution as a joint distribution over response sequence $Y$ and bias sequence $B$.
4. Appendix A is introduced to discuss the derivation of our proposal in equation 7 from Discrete Langevin Proposal [1] in more detail.

We hope that these modifications improve the clarity of our submission.

We would like to briefly highlight the main contributions of our work. Within the field of inference-time controlled text generation (CTG), gradient-based methods offer a flexible and efficient way to enforce constraints on language models. However, these methods typically suffer from a steep tradeoff between fluency and constraint satisfaction. In this paper, we demonstrate that this is a result of the disconnect between the commonly used gradient-based continuous sampling methods and the discrete space of language. To address this, we introduce DAB, a method that leverages gradient information to perform discrete sampling. We demonstrate that our algorithm captures the best aspects of previous state-of-the-art CTG algorithms while enabling a superior balance between fluency and constraint satisfaction, which we demonstrate by beating strong baselines on a range of CTG tasks. Finally, our algorithm is able to achieve these remarkable results while exhibiting superior stability and speed than prior methods. Given the novelty of our method as well as the performance benefits, we believe this is a valuable addition to the field of controlled text generation.

[1] Zhang et al. A Langevin-like Sampler for Discrete Distributions. ICML 2022.

---

### Author Response · Authors · 2024-11-25
**Reminder for Discussion Period**

We would like to thank the reviewers for their valuable feedback on our submission. Given that the last day for the discussion period is approaching, we wanted to provide a reminder in case there are any follow up questions. We are more than happy to address any lingering doubts or concerns.

---

### Meta-Review · Area_Chair_XR11 · 2024-12-22

**Metareview:**

This paper proposed a method for controllable text generation by applying Discrete Langevin Proposal to sample in the discrete space while being able to leverage gradients of the energy function. Experiments on sentiment, toxicity, and lexical control benchmarks demonstrate the effectiveness of the proposed approach.

Strengths:
1. The method outperforms baselines.

Weaknesses:
1. A reviewer pointed out the more modern benchmarks such as jailbreaking language models should be considered, whereas the benchmarks considered in this work are already saturated to some extent.
2. The method is slow due to the autoregressive sampling instead a for loop.

Overall, despite the weaknesses above, overall reviewers seem to like this paper, and I'm recommending acceptance, although I wouldn't mind if the paper gets rejected.

**Additional Comments On Reviewer Discussion:**

Besides to what's mentioned above, a reviewer pointed out that benchmarks considered in this work were already saturated to some extent. Authors clarified that these benchmarks are still good for comparison to baselines. However, I think the reviewer's point does provide constructive feedback --- evaluating this method on more challenging tasks such as jailbreaking a language model would further strengthen this paper.

---

### Decision · Program_Chairs · 2025-01-22

Accept (Poster)